# Temporal controls on silicic acid utilisation along the West Antarctic Peninsula

George E.A. Swann[1,2], Jennifer Pike[3], Melanie J. Leng[1,2,4], Hilary J. Sloane[2,4] & Andrea M. Snelling[4]

The impact of climatic change along the Antarctica Peninsula has been widely debated in light of atmospheric/oceanic warming and increases in glacial melt over the past half century. Particular concern exists over the impact of these changes on marine ecosystems, not only on primary producers but also on higher trophic levels. Here we present a record detailing of the historical controls on the biogeochemical cycling of silicic acid [$Si(OH)_4$] on the west Antarctica Peninsula margin, a region in which the modern phytoplankton environment is constrained by seasonal sea ice. We demonstrate that $Si(OH)_4$ cycling through the Holocene alternates between being primarily regulated by sea ice or glacial discharge from the surrounding grounded ice sheet. With further climate-driven change and melting forecast for the twenty-first century, our findings document the potential for biogeochemical cycling and multi-trophic interactions along the peninsula to be increasingly regulated by glacial discharge, altering food-web interactions.

[1] Centre for Environmental Geochemistry, School of Geography, University of Nottingham, University Park, Nottingham NG7 2RD, UK. [2] Centre for Environmental Geochemistry, British Geological Survey, Nottingham NG12 5GG, UK. [3] School of Earth and Ocean Sciences, Cardiff University, Main Building, Park Place, Cardiff CF10 3AT, UK. [4] NERC Isotope Geosciences Facilities, British Geological Survey, Keyworth, Nottingham NG12 5GG, UK. Correspondence and requests for materials should be addressed to G.E.A.S. (email: george.swann@nottingham.ac.uk).

The west Antarctic Peninsula (WAP) is the most northerly part of Antarctica (Fig. 1) and has experienced the greatest increase in surface atmospheric temperature of anywhere in the Southern Hemisphere during the second half of the twentieth century[1,2]. Although there has been an absence of such warming since the late 1990s[3], the region has continued to experience increases in coastal summer sea surface temperature[4,5] and glacier retreat[6]. These changes have had an impact on both the oceanography[7] of the regional water column as well as phytoplankton communities and higher trophic levels in the marine ecosystem[8,9]. Central to this biological response are diatoms, unicellular siliceous algae, which dominate primary productivity along the WAP with peak abundance focused around spring months[10]. Although diatom productivity is ultimately regulated by light availability, blooms are sustained by intrusions of nutrient rich Upper Circumpolar Deep Water (UCDW) onto the shelf that are mixed into the surface layer via a number of processes, the most significant of which includes brine-induced destabilization of the winter water column following seasonal sea-ice growth[10–12].

Silicon, in the form of silicic acid $[Si(OH)_4]$, represents a key nutrient for diatom growth/uptake[13]. During the biomineralization of silicic acid into particulate hydrous silica, lighter $^{28}Si$ is preferentially incorporated into the frustule over the heavier $^{29}Si$ and $^{30}Si$ with an enrichment factor ($\varepsilon$) in marine systems of $-1.1‰$ to $-1.2‰$ that is independent of temperature, $pCO2_{(aq)}$, iron availability and other vital effects[14–17]. With the progressive uptake of $Si(OH)_4$ increasing $\delta^{30}Si$ ($^{30}Si/^{28}Si$) in both the dissolved and particulate phases, records of diatom silicon isotopes ($\delta^{30}Si_{diatom}$) can be used to examine temporal/spatial changes in photic zone $Si(OH)_4$ utilization, controlled by the biological demand/uptake of $Si(OH)_4$ and the rate at which nutrients are supplied to the photic zone[18–20].

Using the silicon isotope composition of diatom frustules ($\delta^{30}Si_{diatom}$) in a region that is not iron limited[21,22], we document the long-term controls on photic zone $Si(OH)_4$ utilization along the WAP from 12.6 to 0.2 kyr, to assess whether future cycling may be dominated by processes other than sea ice. For example, studies have advocated a role for glacial discharge, wind and solar irradiance in regulating macro-nutrient availability in Antarctic surface waters along the WAP[8,23,24]. In contrast to the modern day, we demonstrate that rates of $Si(OH)_4$ utilization are not closely aligned to changes in sea ice, except from 12.6 to 9.1 kyr and 5.3 to 2.6 kyr. In particular, we show that changes in biogeochemical cycles are highly related to glacial discharge into the photic zone over the last 5.4 kyr, indicating the capability for the system to be driven by different processes as the climate system evolves.

## Results

**Species composition.** Diatoms were extracted from sediments deposited between 12.6 and 0.2 kyr at Palmer Deep, WAP (ODP Site 1098A, 64°51.72′ S, 64°12.47′ W; Fig. 1)[25] and analysed for $\delta^{30}Si$ using a fluorination procedure[26]. Bulk species samples analysed in this study, dominated by *Hyalochaete chaetoceros* spp resting spores (average relative abundance = c. 60%), primarily represent the spring bloom with populations continuing into the summer months[27]. Although *H. chaetoceros* resting spores in their vegetative form are associated with sea ice, they bloom and grow rapidly in well-stratified sea-ice melt water adjacent to the melting ice edge[28]. Although evidence from cultures has pointed towards inter-specific variation in the fractionation factor ($\varepsilon$) for $\delta^{30}Si$[29], questions remain over the extent to which this can be extrapolated to the natural environment[20] with, for example, no evidence of a variable $\varepsilon$ in a core-top study from the Southern Ocean and Antarctic Peninsula[19]. Although no investigation into $\varepsilon$ has been carried out along the surface waters of the WAP, the nearest study to date on a transect from across the Atlantic and Indian sectors of the Southern Ocean south of the Antarctic Circumpolar Current reveals a tightly constrained fractionation factor of $-1.2 \pm 0.1‰$ (ref. 30).

Of relevance to this study are diatoms reproducing in sea-ice brine channels, which have a heavier isotopic composition due to their growth in semi-closed systems[31] and so have the potential to distort sediment records of $\delta^{30}Si_{diatom}$. An important distinction, however, is needed between cryophilic diatoms that predominantly live within or attached to sea-ice and diatoms, which are associated with the sea-ice environment but are not obligate to within or attached to sea ice. Cryophilic diatoms (such as *Amphiprora kjellmanii*, *Nitzschia stellata*, *Nitzschia lecointei*, *Pinnularia quadratarea*, *Pleurosigma* sp. and *Berkleya* sp.) are thinly silicified and are rarely preserved in the sediment record, instead being dissolved within the water column or surface sediments. Within samples analysed here, cryophilic diatoms constitute on average c. 0.6% of the relative abundance of taxa (max = c. 2%) with the remainder not being cryophilic taxa.

*Fragilariopsis curta* and *Fragilariopsis cylindrus* are often called sea-ice diatoms, because they are intimately related to the sea-ice environment and may be found within sea ice[32], so it is apposite to consider these taxa here (average relative abundance = c. 10%). Observations suggest that valves within sea ice are thin[33] and, as such, have low preservation potential. In Antarctic waters, *F. curta* and *F. cylindrus* have been observed as being more prominent in assemblages in open water rather than in ice-covered water[34–36] and, in a series of sediment traps in the Weddell Sea, highest fluxes were found to be associated with ice-free summer phytoplankton blooms following the seasonal retreat of sea ice[32]. As such, sea-ice signals transferred to the sea floor are produced during summer from *F. curta* and *F. cylindrus* blooms that have been seeded from sea ice to the open water[32], hence have been growing and reproducing outside of the sea ice environment. Further, studies have also shown that *F. cylindrus* has very high growth rates ($\sim 0.7$ doublings per day) in the marginal ice zone adjacent to the melting sea-ice edge[28,37]. In summary, the bulk of the fossil diatom record analysed in this study for $\delta^{30}Si_{diatom}$ comprises silica that was synthesized in

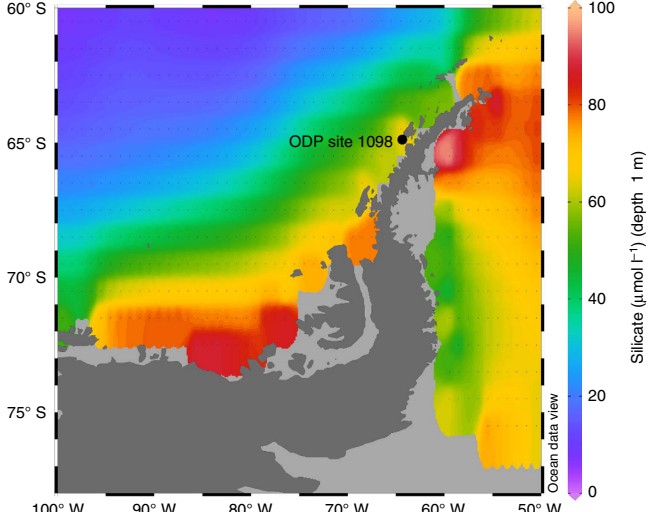

**Figure 1 | Location of ODP site 1098 at Palmer Deep off the West Antarctic Peninsula.** Coloured surface waters indicate silicate concentrations from the World Ocean Atlas 2013 (www.nodc.noaa.gov/OC5/woa13). Map created using Ocean Data View (https://odv.awi.de).

spring waters of the marginal ice zone adjacent to the melting sea ice edge with blooms continuing into the summer months[27]. Accounting for the potentially higher $\delta^{30}Si_{diatom}$ value of cryophilic diatoms (mean = c. 0.6‰; max = c. 2‰) with a simple mass-balance model has a negligible impact (within analytical error) on our $\delta^{30}Si_{diatom}$ data. Our record therefore providing a unique opportunity to examine the factors regulating Holocene nutrient dynamics along the WAP and an indication of how twenty-first century atmospheric/ocean warming and ice-sheet melt might drive further changes in the system.

**Temporal changes in silicon cycling.** At the end of the last glacial through the early Holocene (12.6–8.1 kyr), measurements of $\delta^{30}Si_{diatom}$ are remarkably constant at +0.63 to +0.82‰, with the exception of a notable decrease and increase in $\delta^{30}Si_{diatom}$ at 10.7 and 9.0 kyr, respectively (Fig. 2 and Supplementary Table 1). Following 9.0 kyr, $\delta^{30}Si_{diatom}$ ranges from +1.03 to +0.24‰ with a long-term decline in values from c. +0.9‰ to c. +0.4‰ ($P < 0.001$). Results from the only other $\delta^{30}Si_{diatom}$ record along the coastal Antarctic margin at Adélie Land (East Antarctica)[38] show little similarity to those from the WAP, reflecting differences in source waters, prevailing atmospheric/oceanic conditions and the habitat of diatom taxa (see Supplementary Discussion and Supplementary Fig. 1).

Under an ocean open system model marked by continuous supply of silicic acid to the photic zone, records of $\delta^{30}Si_{diatom}$ can be used to calculate temporal changes in the utilization of $Si(OH)_4$ along the WAP (Fig. 2). Rates of $Si(OH)_4$ utilization

follow $\delta^{30}Si_{diatom}$ with both displaying notable variability after c. 7.1 kyr (Fig. 2), highlighting the potential for nutrient consumption to alter rapidly on centennial/sub-centennial timescales. Hierarchical cluster analysis of the $Si(OH)_4$ utilization data revealed two significant zones (Zone 1: 5.3–0.2 kyr; Zone 2: 12.6–5.6 kyr) (Fig. 3). Whereas rates of $Si(OH)_4$ utilization are close to 50% in the early/mid Holocene period (Zone 2 $Si(OH)_4$ utilization: $\bar{x} = 48.3\%$, interquartile range = 40.3–56.0%) rates are significantly lower ($P < 0.001$) in the late Holocene/ neoglacial (Zone 1 $Si(OH)_4$ utilization: $\bar{x} = 34.2\%$, interquartile range = 17.5–40.9%).

**Early Holocene controls on $Si(OH)_4$ utilization.** For the modern day[39] sea-ice melt plays a key role in regulating spring photic zone productivity and biomineralization by altering water column stability, mixed layer depth, light availability and the suspension of diatoms in the photic zone. However, other mechanisms may have dominated in the past, for example, increases in glacial discharge may have increased the transportation of nutrients, as glacial flour, to the photic zone lowering rates of nutrient usage[23]. Similarly, changes in wind intensity, in particular the southward migration of Southern Hemisphere Westerly Winds (SWWs), can increase the upwelling of UCDW onto the shelf.

To provide further insights into the controls on Holocene $Si(OH)_4$ cycling, significant zones and primary subzones were explored using principal components analysis, to highlight dominant patterns and the inter-relationships between variables (Table 1). Rates of $Si(OH)_4$ utilization were examined alongside:

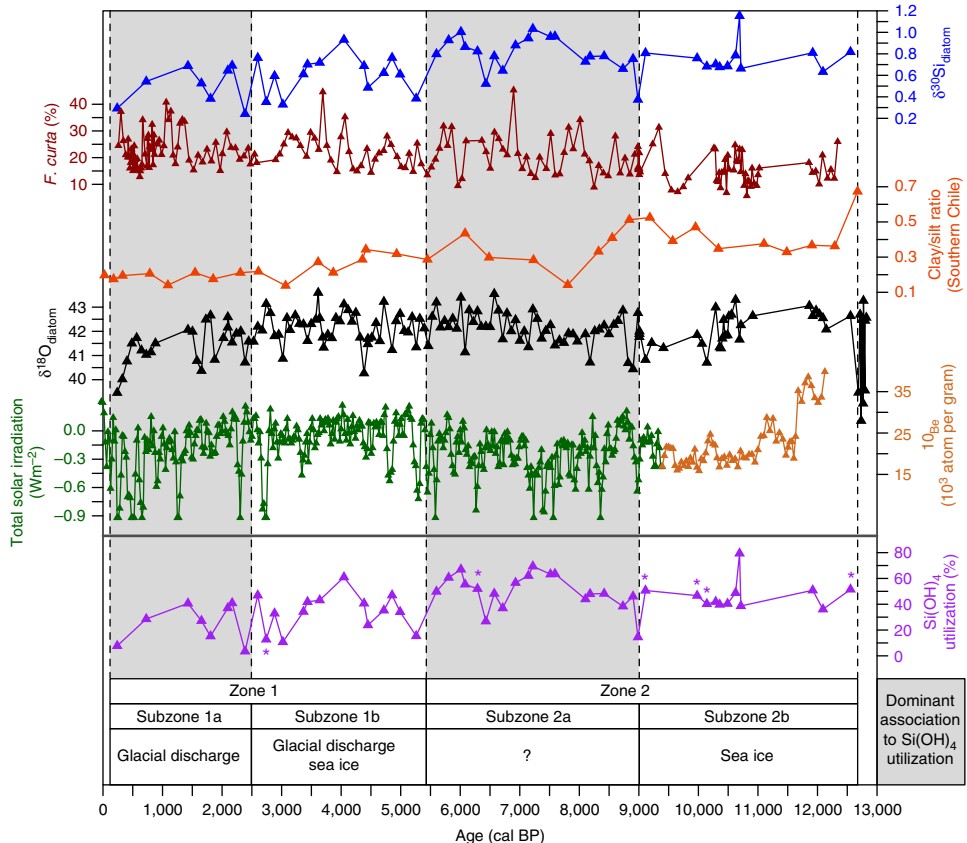

**Figure 2 | Palaeoceanographic records from ODP Site 1098.** Records of $\delta^{30}Si_{diatom}$ (this study) plotted alongside *F. curta* (sea-ice-associated taxa)[40], clay/silt ratios from Southern Chile with higher ratios reflecting stronger Southern Hemisphere Westerly Winds[43] and $\delta^{18}O_{diatom}$ (glacial discharge)[27] alongside TSI for 9.4–0.0 kyr (green line)[44] and Greenland GISP2 ice-core [10]Be for 12.1–9.4 kyr (dark orange line)[45]. Bottom panel displays changes in $Si(OH)_4$ utilization under an open system model. Starred $Si(OH)_4$ utilization data points were removed from any ordination analyses ($n = 6$) (see Methods). Shaded/unshaded intervals indicate transitions between zones/subzones as identified by the hierarchical clustering of samples.

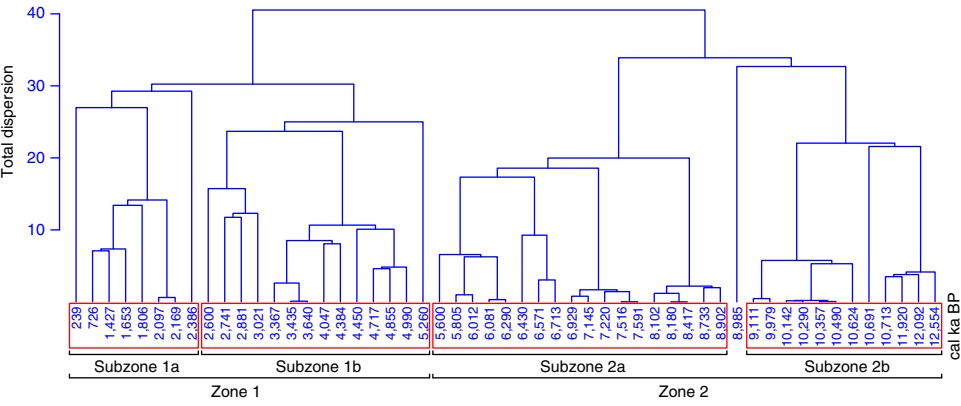

**Figure 3 | Numerical zonation of the $\delta^{30}Si_{diatom}$ data at ODP Site 1098A.** Zone 1 and Zone 2 are statistically significant under a Broken-Stick model.

**Table 1 | Matrix documenting how different environmental variables and processes may potentially increase or decrease rates of Si(OH)$_4$ utilization over the analysed interval at ODP Site 1098A.**

| Environmental variable | ↑ Si(OH)$_4$ utilization | ↓ Si(OH)$_4$ utilization |
|---|---|---|
| ↑ Sea ice (↑ *F. curta*) | ↑ Spring stratification ▶ ↑ productivity | ↑ Brine ▶ ↑ UCDW/nutrient onto shelf |
| ↑ Solar irradiance | ↑ Summer stratification ▶ ↑ productivity | — |
| ↑ SWW | — | ↑ UCDW/nutrient onto shelf |
| ↑ Glacial discharge (↓ $\delta^{18}O_{diatom}$) | ↑ Spring stratification ▶ ↑ productivity | ↑ nutrients (glacial flour) and/or ↑ autumn and winters sea-ice formation ▶ ↑ UCDW/nutrients onto shelf |

SWW, Southern Hemisphere Westerly Wind; UCDW, Upper Circumpolar Deep Water.

(1) changes in seasonal sea-ice abundance, based on the relative abundance of *F. curta* at the same site[40], which is closely associated with pack/fast-ice and commonly found near the sea-ice edge[34,41] with higher abundances, indicating a reduced growing season due to a denser sea ice cover[42]; (2) glacial discharge from the continental ice sheet, as indicated by the oxygen isotope composition of diatom silica ($\delta^{18}O_{diatom}$) measured on the same samples as those analysed here for $\delta^{30}Si_{diatom}$ (ref. 27); (3) changes in the position and intensity of SWWs using clay/silt ratios from the Skyring fjord system of Chile[43]; and (4) measurements of solar irradiance from a combined total solar irradiance (TSI) record for 9.4–0.0 kyr BP (ref. 44) and from the Greenland GISP2 ice-core $^{10}Be$ record for 12.1–9.4 kyr BP (ref. 45). TSI has been widely used to constrain how changes in solar activity drive decadal–centennial scale environmental changes through the Holocene[46,47] and is employed here to understand how solar activity may regulate biogeochemical cycling over multi-centennial timescales.

Previous work from the last deglaciation has advocated a key role for sea-ice in regulating biological processes in the region through the stabilization of the water column during seasonal melting of sea ice[48,49]. Our work advances this by showing that although no single process dominates, sea ice (as indicated by the abundance of sea-ice-associated *F. curta*) is strongly associated with biogeochemical cycling into the early Holocene (subzone 2b: 12.6–9.1 kyr) with enhanced water column stability during spring sea-ice melt retaining diatoms in the photic zone and increasing rates of Si(OH)$_4$ consumption (Figs 4 and 5a). This link between sea ice and Si(OH)$_4$ utilization disappears after 9.1 kyr and into subzone 2a (8.9–5.6 kyr) with rates of Si(OH)$_4$ utilization instead becoming closely related to TSI[44] (adjusted $R^2 = 0.91$, $P < 0.01$, $n = 5$; Fig. 4); Si(OH)$_4$ and TSI did not display an association before this interval.

A link between diatom productivity and solar variability has been proposed for the late Holocene WAP[48]; however, the

mechanisms behind this remain unknown. Although increased TSI can enhance summer stratification of the water column via warming of the surface layer[24,39], leading to increased productivity, the negative correlation between Si(OH)$_4$ utilization and TSI observed here requires an alternative set of interactions. With no comparable analogue in the modern day, we are unable to propose a definitive process that links TSI and nutrient utilization during this interval characterized by reduced glacial discharge following the earlier collapse of the George VI Ice Shelf at 9.6 kyr (ref. 27) and sea ice (*F. curta*) unchanged from before 9.1 kyr.

**Mid/Late Holocene controls on Si(OH)$_4$ utilization.** The Mid/Late Holocene, encompassing the shift from non-cyclic to cyclic internal forcing of the Antarctic climate system and the neoglacial period[27,50], is characterized by a switch away from an association between TSI and rates of Si(OH)$_4$ utilization. Instead, in subzone 1b (5.3–2.6 kyr), a strong positive correlation emerges between Si(OH)$_4$ utilization and both glacial discharge and sea ice (Fig. 4). This switch, advocating a return to sea-ice-driven stabilization of the spring water column in regulating biogeochemical cycling (Fig. 5b), coincides with evidence of reduced winter mixing and UCDW in surface waters at nearby Marguerite Bay[51], a process that would lower nutrient concentrations and cause rates of Si(OH)$_4$ utilization to become more sensitive to inter-annual variations in the strength of spring sea-ice-driven stratification when peak diatom production occurs. The simultaneous emergence of an association between Si(OH)$_4$ utilization and glacial discharge coincides with increases in glacial discharge at the start of the neoglacial related to a strengthened El Niño–Southern Oscillation and enhanced La Niña activity[27]. Rather than glacial discharge regulating water column stability, higher values of $\delta^{18}O_{diatom}$ (less glacial discharge) are linked to increased Si(OH)$_4$ consumption and vice versa (Fig. 4, Table 1).

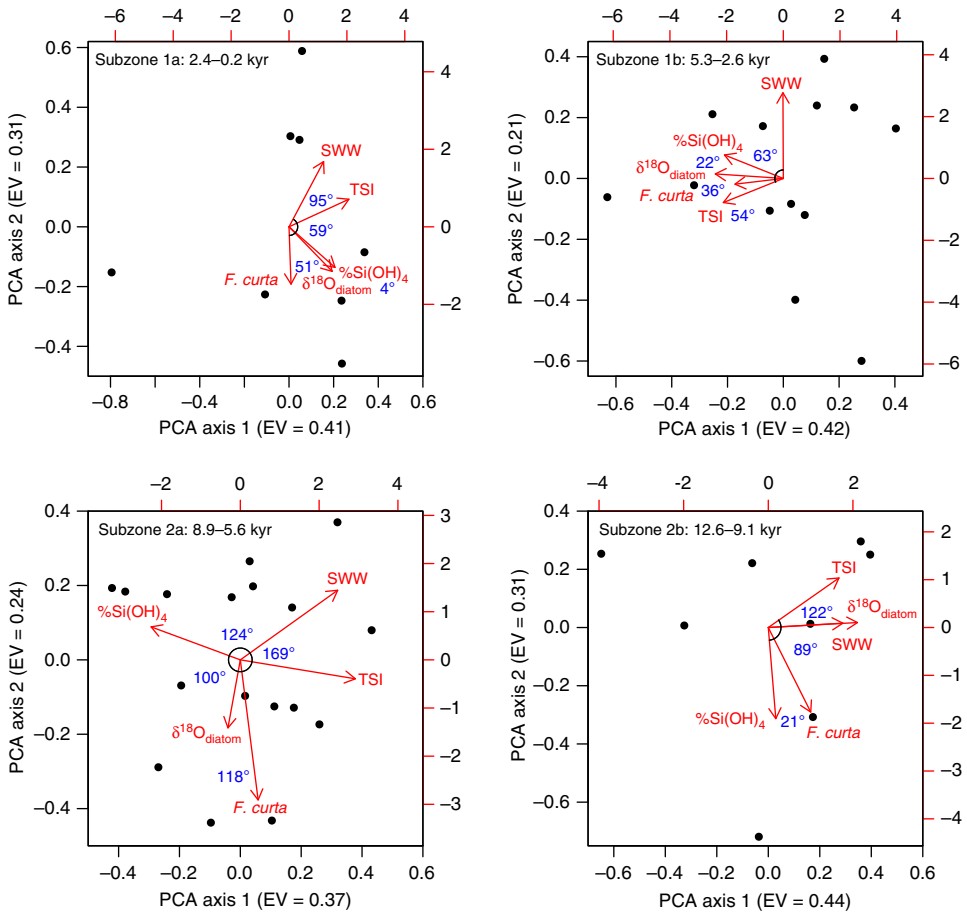

**Figure 4 | Principal component analysis biplots of the Si(OH)$_4$ utilization-defined sub-zones.** Aligned vectors indicate a strong positive correlation between the two variables. Vectors at right angles/opposites indicate no correlation/negative correlation, respectively. Eigenvalues for each axis indicate the variance in the data explained by each axis. SWW, Southern Hemisphere Westerly Winds; TSI, total solar irradiance; %Si(OH)$_4$, percent Si(OH)$_4$ utilization.

Glacial discharge can transport glacially derived nutrients to the photic zone along the WAP[23,52], potentially regulating biogeochemical cycling through this interval alongside sea ice. Although there are few direct measurements of macro-nutrients in Antarctic glacial discharge, records from both Greenland and elsewhere demonstrate the abundance of nutrients including iron in glacial melt[53,54]. Changes in the supply of glacially derived iron are unlikely to induce iron limitation/alter nutrient utilisation along the WAP due to the deep water replenishment of photic zone iron concentrations via winter mixing[52]. Similarly, elevated levels of dissolved silica in meltwater from the Greenland ice-sheet are only observable in fjords and in waters immediately proximal to the ice-sheet with Si(OH)$_4$ rapidly mixed and diluted with marine waters beyond the mouth of individual fjords[55]. Although no comparable data exists for Antarctica, results from northern Marguerite Bay along the WAP indicate that the majority of dissolved silica in the photic zone is supplied from upwelled deep waters[56]. On this evidence, we argue that the link between glacial discharge and biogeochemical cycling through subzone 1b (5.3–2.6 kyr) is not related to the supply of glacially derived nutrients. Instead, we highlight work demonstrating that increased glacial discharge increases subsequent austral autumn and winter sea-ice formation[57], a process that would increase UCDW flow on the shelf through brine-induced destabilization of the winter water column[10–12] and lower nutrient utilization in the following spring bloom. Using this, we argue that periods of high (low) Si(OH)$_4$ utilization in subzone 1b (5.3–2.6 kyr) are

associated with increased (reduced) sea-ice-driven water column stability in spring months and reduced (increased) glacial discharge, which reduces (increases) the flow of nutrient-rich UCDW onto the shelf in winter months (Fig. 5b).

In subzone 1a (2.4–0.2 kyr), the association between sea ice and Si(OH)$_4$ utilization significantly weakens (Fig. 4), a transition that may be related to increased wind strength over the last two millennia[40], which would have limited sea-ice-driven stratification of the water column and lowered diatom concentrations[58,59] (Fig. 2). Instead, in contrast to the modern day, nutrient dynamics over this interval become predominantly associated solely with changes in glacial discharge (Fig. 4) with increased (reduced) Si(OH)$_4$ consumption strongly associated with reduced (increased) glacial discharge. Similar to subzone 1b (5.3–2.6 kyr), a link between rates of Si(OH)$_4$ utilization and the transportation of glacially derived nutrients to ODP Site 1098 is ruled out due to the distance from the grounded ice sheets. Although Si(OH)$_4$ utilization through this period is therefore most likely to be controlled by the aforementioned process in which glacial discharge increases austral autumn/winter sea ice[57] and the flow of nutrient-rich UCDW onto the shelf (Fig. 5c), this argument is seemingly weakened by the absence of a link between sea ice and either glacial discharge or Si(OH)$_4$ utilization in the ordination results (Fig. 4). However, we argue that the absence of any link to sea ice is an artefact driven by the reduction in spring sea-ice-driven stratification over the last two millennia and does not reflect the combined glacial-discharge/sea-ice/UCDW

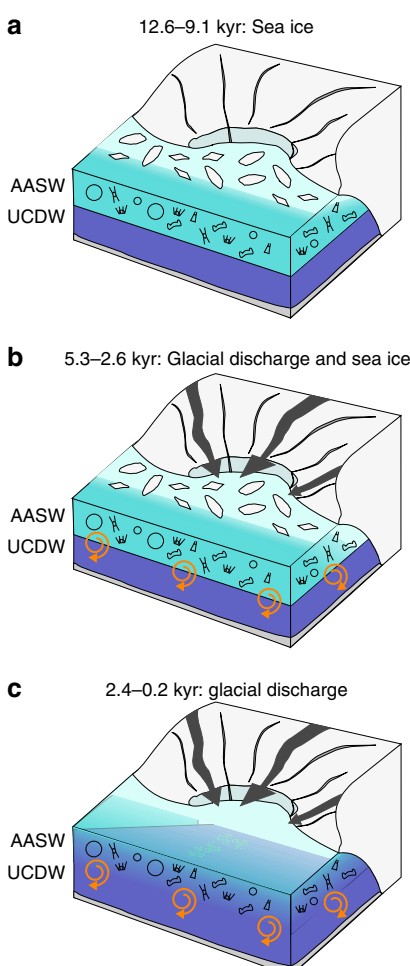

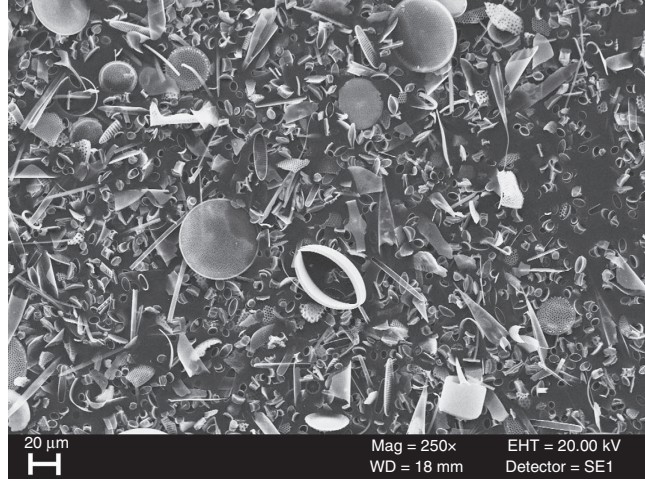

**Figure 6 | Diatom isotope sample at 10.49 kyr.** Scanning electron microscope image of the analysed diatom isotope sample at 10.49 kyr (35.45 mcd), showing the excellent preservation and lack of contamination.

**Figure 5 | Conceptual model of the processes controlling photic zone Si(OH)$_4$ utilization along the WAP.** (**a**) 12.6–9.1 kyr: sea-ice melt enhances water column stability and enables diatoms to remain in the photic zone for longer, increasing rates of Si(OH)$_4$ consumption. (**b**) 5.3–2.6 kyr: water column stability continues to be supported by sea-ice melt. At the same time glacial discharge from the Antarctic ice-sheet regulates the supply of UCDW nutrients to the photic zone through brine-induced destabilisation of the winter water column. (**c**) 2.4–0.2 kyr: glacial discharge from the Antarctic ice-sheet continues to regulate the supply of UCDW-derived nutrients to the photic zone. Dark and light blue indicate UCDW and AASW, respectively, with pale white colour in surface waters reflecting freshwater inputs from sea-ice (**a**,**b**) and glacial discharge (**b**,**c**).

mechanism proposed here in which nutrient flow onto the shelf is aided by brine-induced destabilisation of the winter water column.

**Future implications.** Modern day studies have advocated the role of sea ice in regulating intra-annual to decadal variations in phytoplankton biomass and community structure along the WAP, compounded by changes in cloud cover, wind-driven mixing and glacial melt[8,23,24,39,60]. Our work extends this to demonstrate that the dominant control on biogeochemical cycling varies over time with Si(OH)$_4$ utilization strongly associated to sea ice, glacial discharge and solar forcing at different intervals through the Holocene. Of particular note is evidence that with the exception of the modern day, changes in rates of Si(OH)$_4$ utilization and sea ice are only concordant from 12.6 to 9.1 kyr and 5.3 to 2.6 kyr. With projections indicating further warming and glacial discharge[61] during the twenty-first

century, our results suggest that Si(OH)$_4$ utilization may again become increasingly regulated by glacial discharge and the associated transport of UCDW-derived nutrients to the photic zone.

With primary production along the coastal margin dominated by siliceous organisms, any change in nutrient dynamics has the potential to alter the ecosystem structure and food web interactions. For example, recent reductions in the marginal sea-ice zone in the northern subregion of the WAP have reduced both phytoplankton cell size/net productivity and altered the algal community through relative reductions in the abundance of diatoms, having an impact on both zooplankton and higher trophic levels[8,9]. Increased regulation of biogeochemical cycling through glacial discharge and associated water column stratification may compensate for this by increasing diatom populations and photic zone Si(OH)$_4$ supply, but will not offset ecosystem changes linked to warmer sea surface temperature or loss of sea-ice habitats[8,9]. In addition, further inputs of glacial discharge are likely to exacerbate declines in krill populations linked to lithogenic particles in the water column[62].

Increased transportation of UCDW-derived nutrients to the photic zone and associated reductions in Si(OH)$_4$ utilization will also modify the export of organic carbon both along the WAP[8] and in the open ocean. At locations in the Southern Ocean with negligible sea-ice cover, the availability of Si(OH)$_4$ regulates opal export and so ocean-atmospheric exchanges of $CO_2$ (ref. 63). Although we do not comment on whether our findings advocate a role for the biological pump in regulating Holocene $p$CO$_2$ changes, our records show that rates of silicic acid utilization have reduced through the neoglacial and have the potential to do so again with future increases in glacial discharge. Such changes, if replicated at other coastal sites around West Antarctica and the WAP, would suggest the creation of a pool of under-utilized silicic acid that could alter the local biological carbon pump and stimulate further changes in the open ocean away from the continental margin[20].

## Methods

**Age model.** The age model for ODP Site 1098 follows that published in Pike *et al.*[27] in which previously published down core magnetic susceptibility records[25] and lamina-to-lamina correlations were used to re-evaluate the metres composite depth (mcd) scale for the A and C holes. These were then used against the published particulate organic carbon acccelerator mass spectrometry (AMS) radiocarbon ages for ODP Site 1098 (ref. 64) that were re-calibrated to calendar years using Calib 6.0.2, the Marine09 calibration curve and a 1,230 year reservoir correction.

**Diatom extraction.** Diatoms were extracted and cleaned for isotope analysis using techniques modified for use on coastal Antarctic diatoms[65] with sub-samples previously analysed for $\delta^{18}O_{diatom}$ (ref. 27). Samples were placed in c. 1 ml of 30% $H_2O_2$ at room temperature for $\sim4$ h to disaggregate before being centrifuged in sodium polytungstate three times with progressively lower specific gravities: 2.25, 2.20 and 2.10 g ml$^{-1}$ at 2,500 r.p.m. Extracted material was re-immersed in $H_2O_2$ at 75 °C to remove all organic material adhering to the diatom frustules and left overnight in 5% HCl to dissolve any remaining carbonates.

All samples were checked for purity using scanning electron microscope and ×1,000 magnification light microscopy with these visual analyses, confirming that frustules are exceptionally well preserved and have not been subject to dissolution or other processes that may alter their isotopic composition (Fig. 6). Although we cannot conclusively rule out that dissolution may have affected micro-features on individual frustules, results from sediment traps in Lake Baikal (Russia) demonstrate that such dissolution does not alter $\delta^{30}Si_{diatom}$ (ref. 66). With Lake Baikal experiencing depths of >1,500 m and rates of diatom preservation similar to marine systems (c. 1% of diatoms in Lake Baikal become incorporated into the sediment record[67]), we extend these results to $\delta^{30}Si_{diatom}$ measurements at ODP Site 1098A. Analyses of core-tops in the Southern Ocean have also found little to no effect of dissolution on $\delta^{30}Si_{diatom}$ (ref. 19).

$\delta^{30}Si_{diatom}$ analyses were conducted using a fluorination technique[26] verified through an inter-laboratory calibration exercise[68]. Samples were loaded into nickel reaction vessels and outgassed for 2 h at 250 °C to remove superficial water before reaction with $BrF_5$ for 6 min at 250 °C to remove all –OH bonds. Silicon from the -Si-O-Si layer were then dissociated overnight using an excess of reagent at 550 °C and collected as $SiF_4$. Yield measurements for $\delta^{30}Si_{diatom}$ indicated 100% collection of all silicon. Isotope measurements were made on a Finnigan MAT 253 with values were converted to the NBS28 scale using the NIGL within-run laboratory diatom standard (BFC$_{mod}$) calibrated against NBS28. Replicate analyses of sample material indicate a mean analytical reproducibility ($1\sigma$) of 0.03‰ (range = 0.00–0.07‰, $n = 7$).

**Si(OH)$_4$ utilization.** Using an ocean open system model marked by continuous supply of silicic acid to the photic zone, records of $\delta^{30}Si_{diatom}$ can be expressed as a function of the isotope composition of dissolved silicic acid [$\delta^{30}Si(OH)_4$] supplied to the photic zone, the fraction of Si(OH)$_4$ remaining in the water ($f$) and the enrichment factor between diatoms and dissolved silicic acid ($\varepsilon$):

$$\delta^{30}Si_{diatom} = \delta^{30}Si(OH)_4 + \varepsilon \cdot f \qquad (1)$$

The use of an open model along the WAP is based upon evidence that a closed system model is not appropriate for most oceanic regions[30], including the Southern Ocean[69]. For the region close to the WAP in the Southern Ocean values of $\delta^{30}Si(OH)_4$ and $\varepsilon$ have been constrained at $+1.4‰$ and $-1.2‰$, respectively[30], allowing the calculation of photic zone Si(OH)$_4$ utilization along the WAP:

$$\%Si(OH)_{4utilisation} = 1 - \frac{\delta^{30}Si_{diatom} - 1.4}{-1.2} \qquad (2)$$

Owing to an absence of glacial melt samples/end members, we are unable to account for inputs of glacially derived silicic acid, although contribution of such sources are argued to be minimal at ODP Site 1098 relative to inputs from UCDW due to the distance from the grounded ice sheets.

**Statistical analyses.** To investigate long-term temporal changes in silicon cycling, stratigraphical zones in the Si(OH)$_4$ utilization data set were identified using a hierarchical clustering of a square root-transformed euclidean distance matrix using a Constrained Incremental Sum of Squares agglomeration method[70] using the rioja package within R[71,72]. The significance of individual zones was checked using a Broken-Stick model[73], which assesses whether the amount of variation is greater than that expected for a model with the same number of segments where Pr is the expected proportion of variance for the $k$th zone out of $n$ zones:

$$Pr = \frac{1}{n} \sum_{i=k}^{n} \frac{l}{i} \qquad (3)$$

All ordinations were conducted using the 'stats' and 'vegan' packages in R[71,74]. Diatom counts at ODP Site 1098 were conducted on Core 1098B[40] with dates recalculated based on the age model used in this paper. Linear interpolation of the *F. curta*, SWW and solar irradiance data was then used to obtain values that are comparable to the $\delta^{30}Si_{diatom}$ sample depth/ages. To limit errors, depths where the age difference between the $\delta^{30}Si_{diatom}$ and *F. curta*/solar irradiance data was >50 years were removed from any ordination analyses ($n = 6$). Although the resolution of the clay/silt SWW data required interpolations between samples with greater age differences, the ordinations and subsequent interpretations in this manuscript are similar whether or not the SWW clay/silt ratio data is included. Detrended correspondence analysis with down-weighting was used to determine whether the data exhibited a linear or unimodal response to the latent variables. All zones and subzones produced a first axis gradient length of <1.5, indicating a linear response, and were further explored using principal components analysis with scaling for all variables.

**Data availability.** All $\delta^{30}Si_{diatom}$ and Si(OH)$_4$ utilization data from this manuscript are provided in Supplementary Table 1.

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

## Acknowledgements

This work was supported by the Natural Environment Research Council (grant numbers IP-1234-0511, NE/G004137/1, NE/G004811/1). We thank the staff at the IODP Gulf Coast Core Repository for providing samples and assistance with sampling ODP Site 1098, and three anonymous reviewers who provided valuable and constructive comments on the manuscript. This research was supported by a NERC Isotope Geosciences Facilities Steering Committee (NIGFSC) grant (IP-1234-0511) in addition to Natural Environment Research Council (NERC) grants NE/G004137/1 to M.J.L. and G.E.A.S., and NE/G004811/1 to J.P.

## Author contributions

G.E.A.S., J.P. and M.J.L. conceived the project. H.J.S. and A.M.S. performed the $\delta^{30}Si_{diatom}$ analyses. All authors contributed to interpretations and commented on the manuscript.

## Additional information

**Competing financial interests:** The authors declare no competing financial interests.

