## [Peer Review File · Nature Communications]

Reviewers' comments:

Reviewer #1 (Remarks to the Author):

I'm a Biological Oceanographer/biogeochemist, working on modern marine pelagic ecosystems, without specialized or technical expertise in paleoceanography/paleoclimate. From this perspective, the current WAP ecosystem is of great interest, as it is experiencing rapid climate change and marked ecosystem modifications in response. The plankton system of the WAP, similar to much of coastal Antarctica, is dominated by diatoms and thus the biogeochemical cycling of silica is a critical process. The ms by Swann et al is a fascinating look at recent Holocene Era variations in the factors governing silica cycling and utilization by diatoms. Beside its great intrinsic interest, it also important for helping us understand how the WAP system might respond to future warming, sea ice loss and glacier melting.

I found the modern ecological state of the WAP to be accurately presented, in accord with recent findings and up to date understanding. I also found the results of the investigation into past climate and biogeochemical and ecological variations to be clearly presented in attractive plots, and clearly described in the text.

There are a few things that could be better explained or clarified. The three principal factors governing diatom productivity and silica cycling: solar irradiance, sea ice, and glacier discharge all interact to impact the production system through two proximal mechanisms: in situ light availability and nutrient supply (light vs nutrient limitation). Their effects are clearly distinguished in Figure 3 and depicted nicely in Fig. 4, but it should be clarified that in general, diatom productivity in the modern coastal WAP is mostly regulated by light supply, with nutrient limitation playing a subsidiary role. Like N and P, Si is seldom depleted to below rate-limiting concentrations in the WAP. In some years with large diatom blooms, Si can be fully depleted in the nearshore region. This is less common offshore - even at the core site just 10 miles from Palmer Station. Glacial discharge is cited as a nutrient source, but there are few measurements of N,P and Si in glacial meltwater. Dierseen et al 2002 is often cited in support (as here) but it does not actually contain any nutrient data.

What factors govern solar irradiance (i.e., the solar energy or PAR incident on the surface of the ocean, correct?)? Montes-Hugo showed that cloudiness changed over the 1986-2006 period. Are there other factors operating over the timescale of the data presented here?

The potential role of iron as a limiting nutrient needs to be at least mentioned. Dissolved Fe is relatively abundant in the immediate coastal region, permitting large spring blooms and occasionally, complete Si depletion. Fe concentrations over the shelf are usually lower, and probably limiting, at least under current conditions. There are some reports of iron in glacier runoff.

Reviewer #2 (Remarks to the Author):

This paper presents an interesting data set of silicon isotopic record from West Antarctic Peninsula throughout the Holocene. While not an expert in statistics, I appreciate the effort done to process the data using statistics, an unusual though quite original approach for this type of record.

I would like, however, to express major concerns related to:

- the lack of appropriate reference and discussion of the few previously published papers dealing with Si isotopes in sea-ice diatoms (bearing heavy $\delta^{30}\text{Si}$ signatures) and Holocene record of Southern Ocean (showing different variability as the current WAP record). Considering these

studies would have significantly influenced the interpretation of the results and as such they should be discussed

- the apparent under-estimation of the role of the ACC upwelling and westerlies in supplying nutrients to the Southern Ocean in the statistic processing
- some erroneous / inaccurate claims and shortcomings (e.g. warming inducing higher mixing, reference to silicic acid leakage, discussing open vs. closed model while subsequently ignoring the continental source)

Detailed comments:

55-56 it is an oversimplification, in a context of sea ice environment, to state that Si isotope « reflects changes in photic zone silicic acid utilisation ». It has been shown e.g. by Fripiat et al. (2007) that sea ice diatoms bear a heavier Si isotopic composition by more than + 1permil compared to photic zone diatoms in the mixed layer. Sea ice diatom Si isotope reflects Si utilisation within sea ice. Consequently sediment record is a mixture of both, as discussed in Panizzo et al. (2014). Surprisingly none of these papers (the only ones, to the best of my knowledge) dealing with Si isotopes in Southern Ocean sea ice diatoms and S.O. Holocene records, is referenced. Therefore the interpretation (notably the estimation of Si utilisation, which is the basis of most of the discussion in Fig. 2) of the data is biased by not considering what is available as state of the art on this paper's subject.

When quickly comparing with Panizzo et al. (2014) $\delta^{30}\text{Si}$ sediment record from East Antarctica (off Adelie Land) from the same period in sea-ice zone, it would seem that even if both records are not outright anti-correlated, at least a different number of periods have been identified in Panizzo et al. compared to this manuscript. This should be discussed.

57-59 the authors refer to diatom taxa in summer bloom, but spring bloom can also be significant next to sea ice diatoms. It would be useful to provide stronger evidence to justify neglecting spring and/or sea-ice diatoms.

82-92, Fig. 3 and later. A parameter reflecting upwelling / westerlies wind intensity is absent from the PCA. The upwelling all around the Southern Boundary of the ACC is the primary control of nutrient supply that would then be transported by Ekman to the South in the Sea Ice Zone. It is mentioned only for the early Neoglacial with reference to ENSO. Could e.g. a ENSO index and/or westerlies index be added to the PCA? This highlights a potential gap in the data analysis.

Similarly, it would have been useful to elaborate on the rationale behind the "environmental variables" chosen (fig. 2 and table S2): solar irradiance (TSI), sea-ice and glacial discharge. It is most likely that these three variables are in fact highly inter-related, with a primary control of TSI on the other two. I wonder how this affects the stats and how each variable can be studied independently if one is forcing the two other.

Fig.4. In the legend, main text and scheme I really don't see how, when TSI "warms Antarctic Surface water", it will "increase mixing with the UCDW" (see also my remark on Table S2). Moreover, from the scheme it seems that # of diatoms (and thus productivity?) remains the same throughout the Holocene, is this deliberate?

180-187 The reference to silicic acid leakage for export at low latitude is irrelevant, at least in this version. The supply of Si to low latitude (silicic acid leakage hypothesis in Sarmiento et al. 2004) refers to other water masses and process, namely mostly to the Si:N ratio in subducted water masses SAMW and AAIW north to the polar front, i.e. following to the upper limb of the thermohaline circulation. The study area of this paper, around Antarctica continental margin, looks at the lower limb, i.e. the unused nutrients that will be exported to the deep and bottom ocean (AABW), which has little impact (if any) at low latitude. It may have an impact regionally on the biological carbon pump but probably not on the other basins contrary to what is claimed.

Sup material

67-69: same remark as above: sea ice diatom and their heavy Si isotopic signatures are ignored here.

73-75: the authors previously claimed they use an open system because of Si supply from continent and now they say that the isotopic composition of this source is unknown and thus neglected, this is contradictory. Moreover the open system equation does not work with two different Si sources.

Table S2

I don't understand the mechanisms behind the third column, i.e. why an increase of brine or SST or sea ice formation would increase UCDW and nutrient supply onto shelf.

References

Fripiat, F., Cardinal, D., Tison, J.-L., Worby, A., & André, L. (2007). Diatom-induced silicon isotopic fractionation in Antarctic sea ice. *Journal of Geophysical Research*, 112(G2).

doi:10.1029/2006JG000244

Panizzo, V., Crespin, J., Crosta, X., Shemesh, A., Massé, G., Yam, R., ... Cardinal, D. (2014). Sea ice diatom contributions to Holocene nutrient utilization in East Antarctica. *Paleoceanography*, 29(4), 328-342. doi:10.1002/2014PA002609

Sarmiento, J. L., Gruber, N., Brzezinski, M. A., & Dunne, J. P. (2004). High-latitude controls of thermocline nutrients and low latitude biological productivity. *Nature*, 427, 56-60.

doi:10.1038/nature02204.1.

Reviewer #3 (Remarks to the Author):

I really enjoyed reading this manuscript, which was exceptionally well written (no suggestions for grammatical changes, appropriate referencing, easy to read and understand) and extremely thought provoking - an innovative approach with significant implications. This manuscript presents an exciting new data set - the silicon isotope data from diatoms - used in coordination with several excellent published data sets (% F. curta, diatom concentration, oxygen isotopes on diatoms and total solar irradiance) from the Palmer Deep, with the goal of evaluating controls on primary production in the western Antarctic Peninsula, through the Holocene. The data are analyzed via zonation cluster analysis and then principal component analysis. These tools identify zones, and subzones, within the core, that are interpreted with regard to changes in the dominant control on silica utilization over time - glacial discharge, sea ice, or solar irradiance. Figure 4 presents outstanding simplified block diagrams that illustrate how these controls work; the diagrams are excellent. However, I strongly suggest that Table S2 (matrix documenting how a change in an environmental variable may drive either an increase or decrease in rates of Si(OH)₄ utilisation) be included directly within the paper rather than in the supplement. I know that space limits are considered, however the extra text (and the easy to follow table), in this case, is critical to an understanding of the primary conclusions. Other information contained in the supplement is more methodologic in nature, still critical support for the paper, but not essential in a first read-through.

Here are my comments and questions:

1. Could the authors comment regarding the use of total solar irradiance? I don't disagree with this, but myself have wondered about using TSA versus values that are specific for either high southern latitudes and/or specific seasons, such as spring and summer, as was done by Pike et al. (2013; Dec insolation 60S). I think that this could matter in Table S2 and associated discussion regarding the relationship between total solar irradiance, primary productivity, and silica

utilization. Also, I was unsure of the units for TSI in figure 2 and looked at the Steinhilber et al. paper, which present the data in W/m².

2. I will be honest - I am not intimately familiar with the details of the statistical work presented. Another reviewer should probably be used to evaluate the specifics/validity of the statistics. My assumption is that all the statistical data are valid and that the approach chosen is optimal for this study.

Figure S2 was quite helpful for me, to see how the zonation was developed - and it appears to fit well with the primary data. The principal component bi-plots in figure 3 were also helpful in terms of pulling out relationships among the data sets that are not clearly obvious in figure 2. However, I went back to Taylor and Sjunneskog (2002) to re-familiarize myself with changes in the overall diatom assemblage from the Palmer Deep - and these are similar, but not the same, as the zonations derived in this paper. Any extra information that could be used from the rest of the diatom data? I note that Pike et al. (2013) presented the Palmer Deep % *F. cylindrus* data for example in assisting interpretation of the diatom oxygen isotope data and was interested if any relationships were tested among the new data set and any other species? In particular I wondered about *Chaetoceros* resting spores - since that could be a significant silica sink. I re-read the Pike et al. (2012) supplementary and see the caveats in interpretation for the latest Holocene - so this question may not be directly relevant. One of the things I enjoyed most about this paper is how much it made me think.

Bottom line, my relatively weak understanding of the methods aside, I am comfortable that the applied tools were both necessary and appropriate choices for these datasets. Through the use of these statistics, the authors are able to evaluate which control (sea ice, glacial melting or solar irradiance) is most important.

3. Zones and subzones:

- Lines 114-117: this sentence is not entirely clear - more explanation warranted, along with explanation of block diagram 4A. As noted in comment 1, I suggest greater discussion of solar irradiance - though I know this has been a stumbling block for many researchers. How does warming of AASW increase mixing with UCDW? I don't disagree with the authors, but I remain unclear about the mechanics of this - a few more sentences here would be appreciated.

- Lines 144-146: The wording is confusing to me. In subzone 1a, the authors suggest that controls by sea ice are limited, and instead Si(OH)₄ consumption was related to glacial discharge. So I think what I see is that over this time period, decreased $\delta^{18}\text{O}_{\text{diatom}}$ data, indicated greater glacial discharge and decreased silica consumption. I think this is what they are saying in the next sentence (lines 146-147), but it's presented in a bit of a reversed way in the preceding sentence. The language could be clearer.

4. Future implications:

The final section of the manuscript presents a straightforward rationale as to the significance of the findings of this work, and its oceanographic links beyond the Southern Ocean. The authors do an excellent job here. Given the final section of the manuscript, even with the caveat described (lines 181-182), I think paper is appropriate for Nature Communications.

Our responses to the three reviewers comments are below in bold. Whereas reviewers 1 and 3 requested relatively minor changes, reviewer 2 raised some important issues/questions which we have addressed in full. In responding to these comments we have taken the opportunity to add relevant text to the manuscript/supplementary information. All references mentioned in our reply are cited at the end of this document.

DETAILED RESPONSES

Reviewer 1 (responses in bold):

..it should be clarified that in general, diatom productivity in the modern coastal WAP is mostly regulated by light supply, with nutrient limitation playing a subsidiary role. **This is now made clear in the second paragraph of the manuscript**

Glacial discharge is cited as a nutrient source, but there are few measurements of N,P and Si in glacial meltwater. **The text has been adapted to reflect this point and we have added references to other global studies showing that glacial meltwater can contain high concentrations of both macro and micro-nutrients.**

What factors govern solar irradiance (i.e., the solar energy or PAR incident on the surface of the ocean, correct)? Montes-Hugo showed that cloudiness changed over the 1986-2006 period. Are there other factors operating over the timescale of the data presented here? **Solar irradiance reflects changes in solar activity (Steinheilber et al. 2012) which in turn, as the reviewer suggest, will impact the photosynthetically active radiation (PAR) reaching organisms in the water column. We are aware of the work by Montes-Hugo and others who have shown a link between cloud formation/cover and recent chlorophyll a changes along the west Antarctic Peninsula. However, we are unable to consider these interactions in our Holocene record due to the absence of suitable proxies for cloud processes.**

The potential role of iron as a limiting nutrient needs to be at least mentioned. **We have included some text and added some references to Ardelan et al. (2010) and Huang et al. (2012) showing that the coastal West Antarctica Peninsula region is not iron limited.**

Reviewer 2 (responses in bold):

It is an oversimplification, in a context of sea ice environment, to state that Si isotope « reflects changes in photic zone silicic acid utilisation ». It has been shown e.g. by Fripiat et al. (2007) that sea ice diatoms bear a heavier Si isotopic composition by more than + 1‰ compared to photic zone diatoms in the mixed layer. Sea ice diatom Si isotope reflects Si utilisation within sea ice. Consequently sediment record is a mixture of both... the data is biased by not considering what is available as state of the art on this paper's subject. **We are aware of Fripiat et al. (2007) who shows higher isotopic values for “sea-ice diatoms” due to their growth in closed/semi-closed systems within brine channels. We note, however, that they do not state which taxa were in their analysed samples. An important distinction is needed between cryophilic diatoms which live within or attached to sea-ice (i.e., taxa that were analysed by Fripiat et al. 2007) and other taxa which are associated with the sea-ice environment around Antarctic but are not obligate to within or attached to sea ice. Cryophilic diatoms (e.g., *Amphiprora kjellmanii*, *Nitzschia stellata*, *N. lecointei*, *Pinnularia quadratarea*, *Pleurosigma spp.* and *Berkleya sp.*) are thinly silicified, predominantly recycled within the water column/surface sediments, and are very rarely preserved in the sediment fossil record.**

Within our samples cryophilic diatoms constitute on average c. 0.6% of the relative abundance of taxa (max c. ~2%). The remaining ≥98% of taxa are not cryophilic and so are not affected by the results of Fripiat et al. (2007). Instead the bulk of our samples are derived from *Hyalochaete Chaetoceros* resting spores (average relative abundance ~60%).

These taxa, in their vegetative form, are associated with sea-ice, but bloom and grow rapidly in well-stratified melt water, adjacent to the melting ice edge (Leventer 1998). We acknowledge that *Fragilariopsis curta* and *F. cylindrus* (average relative abundance c. 10%) are often called sea-ice diatoms as they are intimately related to the sea-ice environment and may be found within sea ice, but they are equally prominent in assemblages in the marginal ice zone adjacent to the melting sea ice edge where they have very high growth rates (~0.7 doublings per day for *F. cylindrus*) (Leventer 1998; Sommer, 1989). Hence, the bulk of our analysed diatoms are comprised of silica that was synthesised in waters of the marginal ice zone and not within the brine channels of sea-ice. Accounting for the potentially higher $\delta^{30}\text{Si}_{\text{diatom}}$ value of cryophilic diatoms (mean relative abundance = c. 0.6%; max = c. 2%) with a simple mass-balance model has a negligible (within analytical error) impact on our data. Given this discussion (relevant information has been added to the main text and SI), we contend that our $\delta^{30}\text{Si}_{\text{diatom}}$ signature is reflecting utilisation within the photic zone and not the growth of diatoms in sea-ice.

When quickly comparing with Panizzo et al. (2014) $\delta^{30}\text{Si}$ sediment record from East Antarctica (off Adelie Land) from the same period in sea-ice zone, it would seem that even if both records are not outright anti-correlated, at least a different number of periods have been identified in Panizzo et al. compared to this manuscript. This should be discussed. **The aim of this current paper is to document the long-term controls on photic zone nutrient utilisation along the west Antarctic Peninsula (WAP). As such, there is little purpose in directly comparing our results from the west Antarctic Peninsula (WAP) to those from Adélie Land which is off the East Antarctic coastline on the other side of the continent. Any comparison would require the manuscript to be considerably extended and would dilute the key points we make which specifically relate to the WAP. Whilst we understand the reviewer's point of view, we also strongly believe that more than two Holocene records (ours and Panizzo et al.) are required before any robust attempt can be made to assess spatial variations in biogeochemical cycling across the Antarctic continental margin!**

57-59 the authors refer to diatom taxa in summer bloom, but spring bloom can also be significant next to sea ice diatoms. **The text has been reworded to make it clear that diatoms start blooming in spring surface ocean water and continue to prevail into the summer months.**

82-92, Fig. 3 and later. A parameter reflecting upwelling / westerlies wind intensity is absent from the PCA. The upwelling all around the Southern Boundary of the ACC is the primary control of nutrient supply that would then be transported by Ekman to the South in the Sea Ice Zone... Could e.g. a ENSO index and/or westerlies index be added to the PCA? **We accept that this is a limitation with our manuscript, particularly in light of evidence that wind intensity can be an important factor in the modern system. Accordingly, we have expanded our paper and analyses (including ordinations) to include a proxy for Southern Hemisphere Westerly Winds (SWW) (Lamy et al. 2010). Whilst we believe this improves the manuscript we note that:**

- i. including this data does not changes our main findings or conclusions;
- ii. at no point is SWW associated with changes in $\text{Si}(\text{OH})_4$ utilisation.

Similarly, it would have been useful to elaborate on the rationale behind the "environmental variables" chosen (fig. 2 and table S2): solar irradiance (TSI), sea-ice and glacial discharge. It is most likely that these three variables are in fact highly inter-related, with a primary control of TSI on the other two. I wonder how this affects the stats and how each variable can be studied independently if one is forcing the two other. **The selected environmental variables are justified in relation to the modern day environment in both the introduction and in the discussion. The suggestion that sea-ice and glacial discharge are primarily controlled by TSI is not supported by studies on the modern or palaeo (Holocene) system (Dierssen et al., 2002; Clarke et al., 2008; Montes-Hugo et al., 2009). In addition, the PCA analysis in Figure 3 shows that the environmental variations in individual zones are not correlated.**

Fig.4. In the legend, main text and scheme I really don't see how, when TSI "warms Antarctic Surface water", it will "increase mixing with the UCDW" (see also my remark on Table S2). Moreover, from the scheme it seems that # of diatoms (and thus productivity?) remains the same throughout the Holocene, is this deliberate? **In subzone 2a (8.9-5.6 kyr), having ruled out other possible mechanisms, we can only conclude that TSI must be leading to increased mixing of AASW and UCDW – but we are unable to produce a definitive processes by which this may occur. Rather than “cover up” this issue, we have revised the text to make it explicitly clear that we are being “speculative” and that we are unable to provide an exact mechanism or find an example of such a process in the modern system. With regards to diatom concentrations, this record is distorted by both the large number of *Hyalochaete Chaetoceros* resting spores and dissolution (only c. 2-3% of all frustules are buried in the sediment record). A key advantage of $\delta^{30}\text{Si}_{\text{diatom}}$ is that it is resilient to dissolution and faithfully reflects surface water processes. Consequently, a relationship between sedimentary diatom concentrations and $\delta^{30}\text{Si}_{\text{diatom}}$ should not be expected.**

180-187 The reference to silicic acid leakage for export at low latitude is irrelevant, at least in this version. The supply of Si to low latitude (silicic acid leakage hypothesis in Sarmiento et al. 2004) refers to other water masses and process, namely mostly to the Si:N ratio in subducted water masses SAMW and AAIW north to the polar front, i.e. following to the upper limb of the thermohaline circulation. The study area of this paper, around Antarctica continental margin, looks at the lower limb, i.e. the unused nutrients that will be exported to the deep and bottom ocean (AABW), which has little impact (if any) at low latitude. It may have an impact regionally on the biological carbon pump but probably not on the other basins contrary to what is claimed. **Whilst the reviewer is correct in highlighting that silicon dynamics involving the SAMW and AAIW are more important for silicic acid leakage to low latitudes, the recent paper by Hendry and Brzezinski (2014) cited in our manuscript shows a clear role for nutrients in waters around the margin that end up in AABW. Given that we only raise the possibility for silicic acid leakage to low-latitude in one sentence and do not use this as a key facet of our study, we have not altered the text. We have however, following the reviewers comment, mentioned that changes in nutrient cycling may impact the local biological carbon pump.**

Sup material

67-69: same remark as above: sea ice diatom and their heavy Si isotopic signatures are ignored here. **See response above**

73-75: the authors previously claimed they use an open system because of Si supply from continent and now they say that the isotopic composition of this source is unknown and thus neglected, this is contradictory. Moreover the open system equation does not work with two different Si sources. **As mentioned in the manuscript, our decision to use an open system model is based on contemporary evidence that the region is best represented by such a model (de la Rocha et al. 2011). An open (not closed) system model would also be appropriate for a region dominated by glacial inputs of nutrients. The critical issue here is, as mentioned in the SI and highlighted by the reviewer, the absence of a $\delta^{30}\text{Si}$ end-member for glacially-derived nutrients. A significant amount is unknown about glacial inputs of nutrients ranging from the concentration of different macro/micro-nutrients as well as spatial/temporal variations in this process. Consequently, the issue raised by the reviewer can not be fully resolved. Whilst this therefore introduces some uncertainty into the quantitative estimates of $\text{Si}(\text{OH})_4$ utilisation, the temporal trends in $\delta^{30}\text{Si}_{\text{diatom}}$ and $\text{Si}(\text{OH})_4$ utilisation remain valid. We have expanded Section 1.2 of the SI to cover this point in more detail and to highlight this “caveat” in our study.**

Table S2

I don't understand the mechanisms behind the third column, i.e. why an increase of brine or SST or sea ice formation would increase UCDW and nutrient supply onto shelf. **These mechanisms are explained in the main text of the manuscript. We have copy/pasted relevant sections of this below with the “SST/TSI” point addressed in our response above:**

- “blooms are sustained by intrusions of nutrient rich Upper Circumpolar Deep Water (UCDW) onto the shelf that are mixed into the surface layer via a number of processes, the most significant of which includes brine-induced destabilisation of the winter water column following seasonal sea-ice-growth (Prézelin et al., 2000, 2004; Ducklow et al., 2007).”
- “recent work has demonstrated that glacial discharge increases subsequent austral autumn and winter sea-ice formation (Bintanja et al., 2013), a process that would increase UCDW flow on the shelf (Ducklow et al., 2007) and lower nutrient utilisation in the following spring bloom.”

Reviewer 3 (responses in bold):

I strongly suggest that Table S2 be included directly within the paper. **The Table and text in Section 1.3 of the supplementary information has been moved to the main text of the manuscript.**

Could the authors comment regarding the use of total solar irradiance? **The primary irradiance (TSI) record used in this study is derived from ice-core data from both Greenland and Antarctic (Steinhilber et al., 2012) and documents changes in solar activity which have been shown to drive a number of key decadal-centennial scale changes through the Holocene (e.g., Wanner et al. 2011). Whereas insolation reflects the total amount of energy that has been collected on a surface area within a given time, irradiance is the “instantaneous rate” in which power is delivered to a surface. Whilst insolation (controlled by Milankovitch cycles) is key to understanding long-term climate changes (e.g., glacial-interglacial cycles or long-term Holocene trends), solar irradiance is more appropriate when examining multi-centennial changes (e.g., Wanner et al. 2011) and so is used here when assessing the impact of solar activity in zones through the Holocene. We have added a few lines to the main text to justify our use of the TSI data.**

Also, I was unsure of the units for TSI in figure 2 and looked at the Steinhilber et al. paper, which present the data in W/m². **The units have been added to figure 2.**

I went back to Taylor and Sjunneskog (2002) to re-familiarize myself with changes in the overall diatom assemblage from the Palmer Deep..... [is there] any extra information that could be used from the rest of the diatom data?.... In particular I wondered about *Chaetoceros* resting spores - since that could be a significant silica sink. I re-read the Pike et al. (2012) supplementary and see the caveats in interpretation for the latest Holocene - so this question may not be directly relevant **We have carefully considered this comment and re-examined again the diatom assemblage data from Taylor and Sjunneskog (2002) alongside the caveats mentioned by the reviewer and those raised by Pike et al (2013). In summary:**

- The overwhelming majority of taxa in the diatom assemblages are responding to a variety of environmental conditions (see Taylor and Sjunneskog [2002] and Pike et al. [2013]). This, in addition to the revised age model used in this manuscript and Pike et al. (2013), accounts for the different timing of individual zones in Taylor and Sjunneskog (2003).**
- As the the assemblages data does not simply reflect changes in nutrient utilisation there is no further information which can be extracted from the assemblage data to aid our $\delta^{30}\text{Si}_{\text{diatom}}$ record/interpretation. The potential exception to this are *Hyalochaete Chaetoceros* resting spores (CRS) given their link to spring bloom intensity. However, there is no link between $\delta^{30}\text{Si}_{\text{diatom}}/\text{Si}(\text{OH})_4$ utilisation and CRS abundance reflecting the multitude of environmental processes which ultimately regulate CRS concentrations.**

- Lines 114-117: this sentence is not entirely clear - more explanation warranted, along with explanation of block diagram 4A..... How does warming of AASW increase mixing with UCDW? I

don't disagree with the authors, but I remain unclear about the mechanics of this - a few more sentences here would be appreciated. **See response to reviewer 2 above.**

- Lines 144-146: The wording is confusing to me.... I think this is what they are saying in the next sentence (lines 146-147), but it's presented in a bit of a reversed way in the preceding sentence. **We have altered the text in this section.**

References used in response

Ardelan, M.V., Holm-Hansen, O., Hewes, C.D., Reiss, C.S., Silva, N.S., Dulaiova, H., Steinnes, E., Sakshaug, E. (2010) Natural iron enrichment around the Antarctic Peninsula in the Southern Ocean. *Biogeosciences*, 7, 11-25.

Huang, K., Ducklow, H., Vernet, M., Cassar, N., Bender, M.L. (2012) Export production and its regulating factors in the West Antarctica Peninsula region of the Southern Ocean, *Global Biogeochem Cy.* 26, GB2005, doi:10.1029/2010GB004028.

Leventer, A. (1998) The fate of Antarctic "sea ice diatoms" and their use as paleoenvironmental indicators. In: Lizotte, M., Arrigo, K. (eds). *Antarctic Sea Ice: Biological Processes, Interactions, and Variability*. Antarctic Research Series 73, 121-137 (Washington DC, American Geophysical Union).

Pike, J., Swann, G.E.A., Leng, M.J., Snelling, A.M. (2013) Glacial discharge along the west Antarctic Peninsula during the Holocene. *Nat Geosci.* 6, 199-202.

Sommer, U. (1989) Maximal growth rates of Antarctic phytoplankton: only weak dependence on cell size. *Limnol Oceanogr.* 34, 1109-1112.

Steinhilber, F., Abreu, J.A., Beer, J., Brunner, I., Christl, M., Fischer, H., Heikkilä, U., Kubik, P.W., Mann, M., McCracken, K.G., Miller, H., Miyahara, H., Oerter, H., Wilhelms, F. (2012) 9,400 years of cosmic radiation and solar activity from ice cores and tree rings. *P Natl Acad Sci USA.* 109, 5967-5971.

Taylor, F., Sjunneskog, C. (2002) Postglacial marine diatom record of the Palmer Deep, Antarctic Peninsula (ODP Leg 178, Site 1098) 2. Diatom assemblages. *Paleoceanography* 17, 8001, doi:10.1029/2000PA000564.

Wanner, H., Wanner, O., Grosjean, M., Ritz, S.P., Jetel, M. (2011) Structure and origin of Holocene cold events. *Quaternary Sci Rev.* 30, 3109-3123.

Reviewers' comments:

Reviewer #1 (Remarks to the Author):

The authors have addressed my chief concerns and I find the revised ms to be interesting and important. It links the historical variability in silicic acid utilization to present ecological conditions. This should be of interest to marine and climate scientists outside the immediate scientific area of the research, particularly given the rapid pace of present-day change. The paper is clearly written, illustrated and documented. The authors were very responsive in addressing the additional detailed points I identified in my review.

Reviewer #2 (Remarks to the Author):

I continue to believe this data are valuable and deserve publication and I will not reiterate the points in favour as raised in my first review. However, while the authors did take into account some of my remarks (notably a proxy of westerlies), they still do not provide adequate response to most of my major concerns. I think the data needs more in-depth discussion, with a better description of the proposed hypotheses and their limitations. In this revised version, still, many omissions make the interpretations incomplete and highly speculative (I develop below more particularly why 3 out of the 4 scenarios discussed are most likely invalid) and I'm afraid that this work is simply not suited to fit the length constraints of Nature Communications.

Diatoms living in sea ice

I agree with the authors that the cryophilic species can be neglected if they represent less than 2% with an average of 0.6%. I equally agree that resting spores are dominant and not living in sea ice. The problem is, some species occur in both sea ice and surrounding water. This is why paleo-oceanographers are searching since so long for sea ice proxies that are not species-dependent (like IP25) because species are of little use to determine whether diatoms lived within sea ice or around sea - ice. Unfortunately, *F. curta* and *F. cylindrus* are good examples of this issue, since they can live in these two environments, as acknowledged by the authors in their reply "*Fragilariopsis curta* and *F. cylindrus* (average relative abundance c. 10%) are often called sea-ice diatoms as they are intimately related to the sea-ice environment and may be found within sea ice, but they are equally prominent in assemblages in the marginal ice zone adjacent to the melting sea ice edge". In this same sentence, they claim to have a relative mean abundance of ca. 10% for *F. cylindrus* while in Fig. 2, they provide a record of *F. curta* abundance, which is around 25%. Hence, in total, *F. curta* + *F. cylindrus* would represent 35-40% of diatom abundance. From this important contribution to the total, what part is coming from *F. cylindrus* and *curta* originating from sea ice? The reply is not convincing since they did not consider this *F. cylindrus* and *F. curta* pool but made an estimate of isotopic bias due to the only minor cryophilic diatoms to be max. 2%. This % neglects the species living in both environments that represent 40% of diatoms in their sediment. $\delta^{30}\text{Si}$ signature of diatoms sampled within sea ice are much heavier, whatever the species. So this calculation seems to largely underestimate the potential isotopic bias.

Below, I'm citing extract from L.K. Armand et al. / *Palaeogeography, Palaeoclimatology, Palaeoecology* 223 (2005) 93–126 on *F. curta* : "*Garrison (1991) ascribed Fragilariopsis curta as a species most common in both pack and fast-ice as in previous reports (e.g. Gersonde, 1984; Horner, 1985; Krebs et al., 1987; Garrison et al., 1983, 1987; Garrison and Buck, 1989; Tanimura et al., 1990). The species is also noted in very high abundance in the water column near the sea ice edge (...)*"

Here I'm citing E. Malinverno et al. / *Marine Micropaleontology* 123 (2016) 41–58 "*High concentrations of extant F. curta and especially F. cylindrus are found in fact within sea-ice and in adjacent open waters of the MIZ (Kang and Fryxel, 1992; Cefarelli et al., 2010; Hinz et al., 2012).*" »

In summary, it seems that *F. curta* and *F. cylindrus* can be found equally within sea-ice and in surrounding waters, and in the authors' reply, which focuses on minor cryophilic species, they again choose to ignore this pool (representing 40% of their diatoms) that can bias the $\delta^{30}\text{Si}$ record.

Comparison with $\delta^{30}\text{Si}$ Holocene record in Antarctic sea ice zone.

I strongly disagree with the authors where they reply that "*there is little purpose in directly comparing our results from the west Antarctic Peninsula (WAP) to those from Adélie Land which is off the East Antarctic coastline on the other side of the continent*".

First off, contrary to what they claim, I've never asked them to make a "*robust attempt (...) to assess spatial variations in biogeochemical cycling across the Antarctic continental margin*". I suggested that they should discuss the difference(s) and similarity(ies) with the only previous study using the same proxy on the same Holocene period in the Southern Ocean sea ice zone. At least it's basic scientific practice to inform the reader on state of the art regarding the topic. Much more than that, it could be particularly valuable to compare the two regions, because, yes indeed, WAP is different from East Antarctica - I'm not questioning this, and this is what makes the current study interesting, new,

original and warranting the comparison to highlight the specificity of both Southern Ocean environments.

Use of steady-state model.

I also know and I agree that steady state model is the most common model to be applied in the ocean for Si isotopes and more specifically in the Southern Ocean (SO), but this is generally in the mixed layer. De La Rocha et al. 2011 study cited by the authors is on surface data sampled at one single time. The fact that DLR et al. data are better described by steady state, as are most other mixed layer SO campaigns that measured $\delta^{30}\text{Si}$, does not necessarily imply that the same model can be applied for the whole season or paleoceanography as discussed by Fripiat et al. (Biogeosciences, 9, 2443–2457, 2012). Indeed a seasonal succession of steady state would yield to a Rayleigh at annual time scale (see Fry, 2006). But the issue of Rayleigh vs. Steady state is not even my main worry here. In fact, for any application of steady state model (or for closed system Rayleigh), the equations are valid only if a single Si source is present. The simple maths of steady state model cannot work if there are more than one Si source and/or if the Si concentration and $\delta^{30}\text{Si}$ signature of this single source are not constant. Therefore, it is plain wrong to state that *“A significant amount is unknown about nutrients in glacial melt, ranging from the concentration of different macro/micro-nutrients to the timing of such input and spatial/temporal variations in this process. Whilst the absence of a suitable end-member introduces a degree of uncertainty into the quantification of Si(OH)_4 utilisation through the Holocene, the temporal trends in $\delta^{30}\text{Si}$ and ***Si(OH)4 utilisation remain valid***”*. In the case of significant Si source from glacial in addition to UCDW as discussed in revised version lines 190-200, Equation 2 is not valid and cannot be used to estimate Si utilisation (unless the supply for UCDW is always negligible or the two sources have the same $\delta^{30}\text{Si}$ signatures and Si contents). If the authors consider glacial Si source is significant, then they cannot use equation 2 and certainly cannot claim that it explains the $\delta^{30}\text{Si}$. The alternative might be that the glacial discharge supply other nutrients (e.g. Fe) while Si source remains largely the same (UCDW) but this is not what is written. Therefore scenarios C and D drawn in Fig. 4 are based on this contradiction.

Warming of AASW will "increase mixing with the UCDW"

For starters, I observe this issue was also raised by Rev. 3. The authors now frankly admit they have no clue how warming of surface layer should increase mixing with the water below. I can understand this is indeed difficult to explain since it's simply going against the law of gravity. How can we trust such statement if the authors themselves have no idea to explain the mechanisms behind it? I'm not saying it is not happening: there might very well be an indirect and complex suite of processes, but since this statement is so counter-intuitive, (i) either the interpretation is wrong (which, frankly, is most likely, given the many uncertainties associated to their data processing, cf. above), (ii) either there is no direct / no causal link between AASW T° and mixing with UCDW, and it's the authors' job to propose rational hypotheses (btw what about salinity, winds..?). It is inappropriate to leave the reader thinking s/he should completely reconsider all they knew about basic oceanography. This casts serious doubt on scenario B drawn in Fig. 4.

Reviewer #3 (Remarks to the Author):

Thank you for the chance to take a second look at the manuscript by Swann et al "Temporal controls on silicic acid utilization along the West Antarctic Peninsula." The authors did a thorough job addressing the comments by reviewers, and appreciate that in some instances, the authors are willing to accept that the community simply does not have enough information to provide definitive answers. This is true for example, with questions regarding changes in southwesterly wind strength and direction over time (the topic of many proposals), the absence of data for a del30 Si end member, or speculation regarding silicic acid leakage. I really do appreciate the speculation, as a way to think forward to new projects, and ways to address existing open questions. And the authors provide sufficient text as caveats. I am glad that material from the supplementary information was moved to the main text, particularly new Table 1 and accompanying text. I agree with their comments regarding "sea ice taxa" in terms of the very small contribution of obligate sea ice taxa to sea floor sediments, and clarification of minor points brought up by reviewers 1 and 3.

Our responses to the four points made by Reviewer 2 are below with his/her comments in bold. All references mentioned in our reply are cited at the end of this document.

Diatoms living in sea ice. “I agree with the authors that the cryophilic species can be neglected if they represent less than 2%... I equally agree that resting spores are dominant and not living in sea ice.... The problem is, some species occur in both sea ice and surrounding water... The reply is not convincing since they did not consider this *F. cylindrus* and *F. curta* pool but made an estimate of isotopic bias due to the only minor cryophilic diatoms to be max. 2%. $\delta^{30}\text{Si}$ signature of diatoms sampled within sea ice are much heavier, whatever the species. So this calculation seems to largely underestimate the potential isotopic bias.”

In our initial response and in a revised Section 1.3 of the Supplementary Information we argued that *F. cylindrus* and *F. curta* are mainly associated with the marginal ice zone and are not living within sea-ice. We agree that there is much debate on this subject, as highlighted by the examples the reviewer cites from the literature. We have reviewed and extended our discussion on this topic in the Supplementary Information and continue to emphasise that these taxa along the West Antarctic Peninsula are predominantly associated with the marginal ice zone (Kang and Fryxell 1992, Gersonde and Zielinski 2000, Cremer et al. 2003, von Quillfeldt 2004, Pike et al. 2008). In summary these references detail that:

- whilst *F. curta* and *F. cylindrus* can be found within sea-ice, these individuals are thinly silicified and hence have low preservation potential;
- there is strong observational evidence associating these taxa with ice-free summer blooms;
- highest fluxes in sediment traps from the Weddell Sea are also associated with ice-free summer phytoplankton blooms following the seasonal retreat of sea ice.

We appreciate that this debate may continue, although we note that our view/position is shared by Reviewer 3 in their comments. We believe that the extended Section 1.3 of the Supplementary Information sets out our position and highlights relevant material from the literature to support our argument that *F. cylindrus* and *F. curta* in our samples should not be regarded as cryophilic taxa.

Comparison with $\delta^{30}\text{Si}$ Holocene record in Antarctic sea ice zone: “...I suggested that they should discuss the difference(s) and similarity(ies) with the only previous study using the same proxy on the same Holocene period in the Southern Ocean sea ice zone...”

We now comment on the paper by Panizzo et al (2014) in the “Results” section of the manuscript and direct readers to the Supplementary Information where we have added a new Section 4 and Figure S3 in which this comparison is made.

Use of steady-state model.

In the previous version of the manuscript we proposed two mechanisms by which glacial discharge could be impacting the observed changes in silicic acid utilisation through the late Holocene:

- via inputs of glacially-derived nutrients (Dierssen et al., 2002)
- through a process in which glacial discharge increases autumn/winter sea-ice formation (Bintanja et al., 2013) which would alter the brine-induced destabilisation of the winter water column and control the flow of UCDW and nutrients onto the shelf (Prézelin et al., 2000, 2004; Ducklow et al., 2007).

Whilst we introduced both processes in our previous version of the manuscript, we advocated the “glacially-derived nutrients” explanation. This generated the concerns raised by the Reviewer who (correctly) pointed out the problems this causes. The recent publication of two papers on silicic acid concentrations in glacial meltwater (Meire et al., 2016) and on $\delta^{30}\text{Si}$ systematics in Marguerite Bay along the West Antarctic Peninsula (Annett et al. In Press) have allowed us to review this position. In particular these papers indicate that:

- inputs of glacially-derived nutrients are rapidly mixed into the ocean. As such, elevated silicic acid concentrations only impact locations that are immediately proximal to the ice-sheet;
- the vast majority of photic zone silicic acid along the WAP originates from UCDW.

Accordingly, it becomes highly unlikely that inputs of glacially-derived nutrients would have reached ODP Site 1098 through the late Holocene (due to the distance from the core site to the grounded ice-sheets) and suggests that our focus on the “glacially-derived nutrients” was incorrect. The above discussion is included within our revised manuscript and we now propose that the link between glacial discharge and silicic acid utilisation is through autumn/winter sea-ice formation and subsequent brine-induced destabilisation of the winter water column. As we now argue that silicic acid is only supplied by UCDW, we can safely use a steady-state model and the concerns raised by Reviewer 2 on this issue are no longer relevant – although we thank the reviewer for the time he/she spent considering this point.

Warming of AASW [through solar irradiance] will "increase mixing with the UCDW".

We have removed all mention of this from the text and deleted the relevant panel from Figure 4.

References

- Annett, A.L., Henley, S.F., Venables, H.J., Meredith, M.P., Clarke, A., Ganeshram, R.S. (In Press) Silica cycling and isotopic composition in northern Marguerite Bay on the rapidly-warming western Antarctic Peninsula. *Deep-Sea Res PT II*. doi:10.1016/j.dsr2.2016.09.006.
- Bintanja, R., van Oldenborgh, G.J., Drijfhout, S.S., Wouters, B. Katsman, C.A. (2013). Important role for ocean warming and increased ice-shelf melt in Antarctic sea-ice expansion. *Nat Geosci*. 6, 376-379.
- Cremer, H., Roberts, D., McMinn, A., Gore, D., Melles, M. (2003) The Holocene diatom flora of marine bays in the Windmill Islands, east Antarctica. *Bot. Mar.* 46, 82-106.
- Ducklow, H.W., Baker, K., Martinson, D.G., Quetin, L.B., Ross, R.M., Smith, R.C., Stammerjohn, S.E., Vernet, M., Fraser, W. (2007). Marine pelagic ecosystems: the West Antarctic Peninsula. *Phil. Trans. R. Soc. B*. 362, 67-94.
- Gersonde, R. and Zielinski, U. (2000) The reconstruction of late Quaternary Antarctic sea-ice distribution-the use of diatoms as a proxy for sea-ice. *Palaeogeogr. Palaeoclimatol. Palaeoecol.* 162, 263–286.
- Kang, S.-H. and Fryxell, G.A. (1992) *Fragilariopsis cylindrus* (Grunow) Krieger – the most abundant diatom in water column assemblages of Antarctic marginal ice-edge zones. *Polar Biol.* 12, 609-627.
- Meire, L., Meire, P., Struyf, E., Krawczyk, D.W., Arendt, K.E., Yde, J.C., Pedersen, T.J., Hopwood, M.J., Rysgaard, S., Meysman, F.J.R. (2016) High export of dissolved silica from the Greenland Ice Sheet. *Geophys Res Lett.* 43, 9173-9182.
- Panizzo, V., Crespin, J., Crosta, X., Shemesh, A., Massé, G., Yam, R., Mattioli N., Cardinal, D. (2014) Sea ice diatom contributions to Holocene nutrient utilization in East Antarctica. *Paleoceanography* 29, 328-342.
- Pike, J., Allen, C.S., Leventer, A., Stickley, C.E., Pudsey, C.J. (2008) Comparison of contemporary and fossil diatom assemblages from the western Antarctic Peninsula shelf. *Mar Micropaleontol.* 67, 274-287.
- Prézelin, B.B., Hofmann, E.E., Mengelt, C., Klinck, J.M. (2000) The linkage between Upper Circumpolar Deep Water (UCDW) and phytoplankton assemblages on the west Antarctic Peninsula continental shelf. *J Mar Res.* 58, 165-202.
- Prézelin, B.B., Hofmann, E.E., Moline, M., Klinck, J.M. (2004) Physical forcing of phytoplankton community structure and primary production in continental shelf waters of the Western Antarctic Peninsula. *J Mar Res.* 62, 419-460.
- Von Quillfeldt, C.H. (2004) The diatom *Fragilariopsis cylindrus* and its potential as an indicator species for cold water rather than for sea ice. *Vie Milieu* 54, 137-143.

REVIEWERS' COMMENTS:

Reviewer #2 (Remarks to the Author):

This is the second revised version. In the previous version, I raised 4 points that were still not correctly addressed.

- sea-ice species. The authors have now developed their view and I agree their hypothesis might be valid.

- comparison with similar work (Panizzo et al.). The minimum required is now done (i.e. citation in the main text of this previous work and direct comparison of the 2 dataset in the supplementary material).

- use of steady state isotopic model with 2 Si sources. Since now the authors removed one of the source (which was highly controversial / unlikely) and found some recent / in press reference to support this sudden change of their view of the WAP system, I agree steady-state is adequate, at least with regards to basic maths.

- Warming of AASW [through solar irradiance] will "increase mixing with the UCDW". The authors completely removed this weird assumption. Fine. Still they do not provide explanation on the relationship they have in their data.

In its current stage, the paper is more scientifically correct and balanced. I still believe that it would have benefited to be published in a longer version (unsuitable for Nature Communications) because the data are indeed very interesting but without unequivocal interpretations / conclusions. I leave this aspect with the editor's decision.

Reviewer 2 did not have any further comments/suggestions for improving the manuscript.

All editorial comments/requests were made in full including the transfer, where appropriate, or material from the Supplementary Information to the main text.